# CMV and HIV Coinfection in Women from a Region in Eastern Europe

**DOI:** 10.3390/jpm13111539

**Published:** 2023-10-26

**Authors:** Stela Halichidis, Mariana Aschie, Georgeta Camelia Cozaru, Mihaela Manea, Nicolae Dobrin, Sabina E. Vlad, Elena Matei, Gabriela Izabela Baltatescu, Anca Florentina Mitroi, Mihai Catalin Rosu, Bogdan Florentin Nitu, Ghiulendan Resul, Anca Antonela Nicolau, Ana Maria Cretu, Anca Chisoi

**Affiliations:** 1Clinical Hospital of Infectious Diseases, 100 Ferdinand Blvd., 900178 Constanta, Romania; shalichidis@yahoo.com (S.H.); laborator@infectioaseconstanta.ro (B.F.N.); contact@infectioaseconstanta.ro (G.R.); 2Faculty of Medicine, Ovidius University of Constanta, 1 Universitatii Street, 900470 Constanta, Romania; mariana.aschie@univ-ovidius.ro (M.A.); nicu.dobrin@365.univ-ovidius.ro (N.D.); 3Sf. Apostol Andrei Emergency County Hospital, 145 Tomis Blvd., 900591 Constanta, Romania; georgiana.cozaru@365.univ-ovidius.ro (G.C.C.); gabriela.baltatescu@univ-ovidius.ro (G.I.B.); anca.mitroi@365.univ-ovidius.ro (A.F.M.); antonela.nicolau@365.univ-ovidius.ro (A.A.N.); cretu_anamaria@yahoo.com (A.M.C.);; 4Center for Research and Development of the Morphological and Genetic Studies of Malignant Pathology, Ovidius University of Constanta, 145 Tomis Blvd., 900591 Constanta, Romania; mihaela.manea@sunmedical.ro (M.M.); elena_matei@365.univ-ovidius.ro (E.M.); mihai.rosu@365.univ-ovidius.ro (M.C.R.); 5SC ProDiagnostic SRL, 30 Farului Street, 90060 Constanta, Romania

**Keywords:** with human immunodeficiency virus, human cytomegalovirus, coinfection, CD4, IgG, antiretroviral therapy

## Abstract

(1) Background: Human cytomegalovirus (CMV) infection is one of the most frequent opportunistic infections in immunosuppressed patients. Romania has one of the highest incidences of patients living with human immunodeficiency virus (HIV) which determines an immunosuppressive state. The aim of this study was to establish the prevalence of CMV infection among women living with HIV in Southeastern Romania and also to evaluate and correlate antiretroviral therapy (ART) with CD4 level and CMV disease evolution. (2) Methods: Seventy women living with HIV from Southeastern Romania were screened for CMV infection using antigen quantification. Of these, 50 were included in the study. First, the patients filled out a questionnaire regarding social conditions and other associated diseases. Then, we explored the statistical correlations between the data and HIV status, CD4+ cell counts, viral load, and antiretroviral therapy (ART). (3) Results: Median age of the patients was 33 years. Twenty-nine cases were diagnosed with HIV after sexual life beginning and 21 before. Most of the patients had a CD4 level over 200 cells/µL. ART duration in the CD4 under 200 cells/µL group was a bit longer than that in the CD4 over 200 cells/µL group. Forty-one patients had undetectable viremia. CD4 average value in the lot of patients with undetectable viremia was 704.71 cells/µL and in the lot with detectable viremia was 452.44 cells/µL. Viremia values correlated negatively with CD4 level. A positive correlation between IgG CMV values and ART therapy length was identified. A negative significant correlation between values of IgG CMV and values of CD4 was identified. CD4 value correlated negatively with IgG CMV values and with CMV avidity. (4) Conclusions: IgG CMV values had a weak positive correlation with ART therapy length, and a negative statistically significant correlation with values of CD4. CMV avidity has a negative correlation with CD4 value.

## 1. Introduction

Human cytomegalovirus (CMV) is a ubiquitous agent that can cause infection at any time during the course of life. It infects individuals from diverse geographical and socio-economic backgrounds. The human cytomegalovirus (CMV) is a beta-herpes virus and is considered an important viral opportunistic pathogen in patients with HIV [1].

Infections by CMV continue to be an important health problem in certain patient populations, such as newborns, recipients of solid organs or bone marrow and acquired immunodeficiency syndrome (AIDS) patients. In these groups, CMV is a major cause of morbidity and mortality. The risk of exposure to CMV increases with age [2].

Seroprevalence of CMV infection in people living with HIV (PLWH) is also high, ranging from 50% to above 90% [3,4]. Detectable CMV viral loads are associated with increased mortality, even in the era of highly active antiretroviral therapy (HAART) [5,6].

In an United States study black participants had a 37% higher risk of CMV exposure as compared to non-Hispanic white participants. In same study Mexican American participants had a 93% higher chance of contracting CMV than non-Hispanic white individuals [3]. The seroprevalence of CMV infection in Iranian HIV-positive patients was 94%. CMV infection prevalence is higher in patients with low socioeconomic status and low level of education. The maximum prevalence was detected in patients with unsafe sex and IDUs. A low prevalence of CMV infection was identified in patients with other routes of HIV transmission, iatrogenic or vertical transmission [4]. In people living with HIV (PLWH) category, CMV causes various life-threatening infections like retinitis, pneumonia, encephalitis or enteritis. These infections last their entire life and can antagonize the immune system. The effect is more important among PLWH who are not on ART than those on ART [5].

Neurological involvement is less common and can manifest as encephalitis, myelitis, polyradiculopathy, and mononeuritis multiplex. In the pre-ART era, up to 40% of HIV-infected patients had advanced disease [7].

The mortality rates in PLWH are directly linked to CMV viremia. CMV infection is a cofactor in AIDS progression [6].

As with other herpes viruses, CMV remains latent in the infected cells throughout life and in immunocompromised individuals it reactivates to cause clinical illness [2]. Most cases of CMV disease occur in PLWH with advanced immunosuppression. The patients with HIV and cluster of differentiation 4 (CD4) T-cell counts <200 cells/µL are at significant risk for CMV reactivation, leading to invasive disease. CD4 values under 200 cells/µL are also a risk for other opportunistic infections, not only for the reactivation of viruses from each individual’s pool.

The aim of this study was to establish the prevalence of CMV infection among women living with HIV in Southeastern Romania, a region in Eastern Europe, and to evaluate and correlate antiretroviral therapy (ART) with CD4 level and CMV disease evolution.

## 2. Materials and Methods

### 2.1. Participants, Data Collection and Ethical Statement

This study contains preliminary results of a larger research which includes investigation of CMV shedding in vaginal secretion. Seventy women living with HIV from Southeastern Romania (Dobrogea Region, Constanta County) were screened for Human cytomegalovirus immunoglobulin G (IgG CMV) and its avidity levels (blood values). Samples were collected from the Daily Clinic files of women living with HIV under surveillance of Clinical Infectious Diseases Hospital from Constanta.

A standardized questionnaire was used to gather demographic data. Parameters identified and assessed in the questionnaire were time of first sexual intercourse, number of sexual partners, use and type of birth control, and self-reported history of STI infections. Medical history was retrospectively retrieved from hospital medical records.

After collection of serum samples and demographic data, samples were tested for CMV.

Only 50 patients meet the principal inclusion criteria of the study (IgG CMV—seropositive). From blood samples, the quantity of specific CMV antibodies IgG and the avidity, viral load, and CD4+ cell counts were dosed. The data obtained were then analyzed by correlation with data on HIV status, antiretroviral therapy (ART) and other parameters using statistical methods.

This study was approved for human subjects work by the Clinical Infectious Diseases Hospital Constanta and by the “Ovidius” University of Constanta. Participation of subjects was entirely voluntary and written consent was required for inclusion in the study. Informed consent was obtained from all subjects involved in the study. All personal identifiers were removed from samples to ensure patient confidentiality.

### 2.2. Specimen Collection, HIV and CMV Serology

Blood samples were collected by venous puncture and processed according to probe preparation for each type of determination.

IgG for CMV was measured using the Dia Source CMV Ig G kit (with a specificity of 98.9% and sensitivity of 99.5%), which is intended for the detection of specific IgG antibodies in a sample by means of a sandwich type of the ELISA method (i.e., a solid phase coated with specific antigen—antibody from the analyzed sample—labelled antibody) with ADALTIS Analyzer—GEN—4 according to the kits’ manufacturer’s instructions. The labelled antibody (conjugate) is an animal immunoglobulin fraction to human IgG conjugated with horseradish peroxidase. Peroxidase activity is determined in the test by a substrate containing TMB. Positivity is indicated when blue color appears; after stopping solution has been added, blue changes to yellow. The yellow color intensity is measured by a photometer at 450 nm, and it is proportional to the concentration of specific IgG antibodies in the sample.

Before determination of IgG CMV, the serum was extracted from each blood sample by centrifugation at 3000× *g* within 30 min using Thermo Scientific SL16R centrifuge. The samples were diluted 1:101 with Sample Dilution Buffer before testing. Ninety-six-well microplates coated with capturing antibodies were used: anti-CMV Ig G antibodies for quantitative determination, and for avidity by Index of Avidity, we used the same kit with Avidity Solution.

HIV viral load was measured using Reverse Transcriptase–Polymerase Chain Reaction (RT-PCR) method using 7500 Fast Real TIME Systems (Applied Biosystems, Waltham, MA, USA) and Xpert HIV-1 Viral Load test reagents (Cepheid AB, Solna, Sweden) as described previously [8].

CD4 level was determined through lymphocytic immunophenotyping using flow cytometry with acoustic focusing cytometer (AB Applied Biosystems Attune, Waltham, MA, USA), and reagents from AQUIOS Tetra-1 Panel (Beckman Coulter, Brea, CA, USA), as described previously [8].

### 2.3. Statistical Analyses

All the statistical analyses were computed using SPSS Statistics (IBM SPSS Statistics for Windows, Version 23.0). The descriptive statistics (mean, range and percentage) were computed for continuous variables (age and antiretroviral therapy). After applying the normality Shapiro–Wilk test, we have chosen the Chi-square test for categorical variables as inferential statistics. All correlations are nonparametric (Spearman) and used only values from a specified time point. Women living with HIV, aged between 18 and 70 years old, sexually active and signed informed consent were included in the study. We excluded women with no IgG CMV detected cases with antiretroviral therapy less than 6 months and absence of informed consent. After inclusion–exclusion criteria we obtained a group of 50 representative cases.

## 3. Results

### 3.1. Socio-Demographic Characterization of the Patients

The median age of the patients (50 PLWH, IgG CMV-seropositive women) was 33.32 (SD = 8.25) with no considerable differences between median age of the rural-area patients (33.44 with SD = 5.35) and urban-area patients (33.25 with SD = 9.58).

From the standard deviation (SD) of the two sub-groups, it may be assumed that patients from the urban area are more heterogenic in age. The youngest patient at the study moment was 18 years old (urban area, HIV diagnosed in childhood at 7 years old, first intercourse 15–18 years old, 2–6 sexual partners declared) and the oldest 70 years old (urban area, HIV diagnosed in adulthood, first intercourse after 19 years age, only one sexual partner declared).

From 50 patients included in the study, 32 patients were from urban areas and 18 from rural areas (Table 1). From 18 cases from the rural area, 6% had their first intercourse before turning 14 years old, 64% between 15 and 18 years old and 28% after 19 years old. Similar percentages were identified for cases from urban areas (6% before 14 years old, 41% between 15 and 18 years old, 53% after 19 years old age) (Figure 1). Regarding the number of sexual partners, patients from the rural area declared they had only one sexual partner in a larger percent (28%) than those from the urban area (16%). In the category of 2–6 sexual partners declared, in rural and in urban area patients, percentages were close (72% for patients from rural area, 81% for those from urban area) and represented the vast majority of declared number of sexual partners. Only one patient (from the urban area) declared more than six sexual partners (Figure 1). These data were self-declared.

As for the ways of acquiring HIV, defined from the age at HIV diagnostic and self-declared information from the patient, the diagnosis occurred after the beginning of the sexual life in 29 cases (considered as sexual transmitted HIV), and before the beginning of the sexual life in 21 cases (Table 1). Percentages of the two categories are close (parenteral or trans-placental acquisition of HIV represents 42% of cases, and sexual acquisition represents 58% of cases). The youngest age at HIV diagnosis was 3 years old. The oldest age at HIV diagnosis was 63 years old (the patient is from the urban area, with one declared sexual partner and beginning of sexual life after age 19). Median age at HIV diagnosis is 18.82 years. In the rural area, the median age was 19.61 years (SD = 9.88) and, in the urban area, it was 18.38 years (SD = 12.91). Differences between the two areas are not notable. The only significant difference is found in standard deviation values which may indicate a heterogeneity in the urban area.

Demographic characteristics did not correlate with HIV infection period length (r(48) = 0.020, *p* = 0.889) and do not influence the age at HIV diagnostic (r(48) = −0.071, *p* = 0.625). HIV infection period length did not have notable differences between the rural or urban area (rural 13.83 years, and urban 14.88 years), as well as ART duration (rural 13.22 years, urban 12.88 years). Comparing the SD of HIV infection duration (rural area SD = 6.74, urban area SD = 8.83), it may be concluded that patients from the urban area were more reticent or uncompliant to treatment than those from the rural area (Figure 2).

### 3.2. Correlations between ART, HIV Viremia and CD4 Level of the Patients

In this study, only patients with an ART duration more than 6 months were included. The shortest ART period was 1 year and the longest 22 years. CD4 level was over 200 cells/µL for most of the patients taken in study (Table 1). Only three patients had CD4 values under or equal to 200 cells/µL. All three patients had undetectable viremia (under 50 copies/mL) and ART for more than 15 years. Two of the three patients with CD4 under 200 and undetectable viremia levels had HBV infection associated with no major health issues. The third patient had no other complication or infectious disease associated other than HIV and CMV. Their general health state was good at the moment of the screening.

Average CD4 value in studied cases was 659.3 cells/µL (SD = 318.71).

ART duration in the CD4 under or equal 200 cells/µL group had an average of 16.67 years (SD = 2.56) and in the CD4 over 200 cells/µL group had an average of 12.77 years (SD = 6.43). In the CD4 under or equal 200 cells/µL group, there were only three patients, the rest of them (47 patients) being included in the CD4 over 200 cells/µL group. CD4 level did not correlate with ART duration (*p* = 0.384).

The CD4 average value in the patient lot with undetectable viremia was 704.71 cells/µL (SD = 331.13) and in the detectable viremia lot was 452.44 cells/µL (SD = 123.34). Looking at the SD values in undetectable and detectable viremia groups, CD4 values in the detectable viremia group seem less heterogenic than those in the undetectable viremia group. Viremia values correlated negatively with CD4 level (r(48) = −0.393, *p* = 0.005), as expected (Table 2).

Most patients (41 cases) in the study had undetectable viremia (less than 50 copies/mL) (Table 1). Only 9 patients out of 50 had detectable viremia (more than 50 copies/mL).

The median ART duration in patients with undetectable viremia was 13.1 years (SD = 6.29). The Median ART duration in cases with detectable viremia was 12.22 (SD = 6.78). The level of viremia did not correlate with ART duration (*p* = 0.632).

### 3.3. IgG CMV, CMV Avidity, HIV Viremia and CD4 Values Correlations in the Studied Lot

The presence of CMV infection was a study admission condition.

Median value of IgG CMV in the studied lot was 215.71 U/mL. The standard deviation value was 71.29 (Table 1). The lowest IgG CMV value was 30.7 U/mL in a patient with detectable HIV viremia (75,300 copies/mL), and low avidity for CMV (<40% avidity) and with CD4 over 200 cells/µL. The patient was from an urban area, aged 3 at HIV diagnosis, with 19 years of ART. Due to the lack of IgM values, we cannot sustain the idea of newly acquired CMV infection.

In patients with undetectable viremia, the median value for IgG CMV was 216.55 copies/µL (SD = 71.25). In the group of detectable viremia, the patient’s median value of IgG CMV was 211.88 (SD = 75.64). The median values are similar in the two groups.

Comparing IgG CMV values with ART therapy length identified a positive correlation between the two variables (r(48) = 0.024) and no statistical correlation (Table 2). The IgG CMV maximum value in our lot was 283.7 U/mL. The patient had undetectable HIV viremia, 14 years of ART and is 32 years old, acquiring HIV after starting sexual life. Immunodepression in HIV-positive patients has many mechanisms. ART is a therapy dedicated to HIV and not to CMV infection. Immunoglobulins G type are indicators of protection against disease. The longer ART time, the greater possibility of seroconversion and rising of IgG levels. Also, the other immune mechanism involved in reaction to CMV infection may produce rising of IgG CMV over time. This theory on the positive correlation between IgG CMV values and ART length shows the importance of treating CMV independently from HIV therapy in HIV-CMV coinfected patients.

If comparing the IgG CMV median value according to CD4 level (over or under 200 cells/µL) in the group of CD4 ≤ 200, the median IgG CMV value was 191.60 U/mL (SD = 123.93) and in the group with CD4 > 200, the median value of IgG CMV was 217.25 U/mL (SD = 69.29). Even the median value does not show great differences between two groups; SD values show a heterogeneity in the measured values of IgG CMV in the group of patients with CD4 ≤ 200 cells/µL. By applying the Spearman test to compare values of IgG CMV with values of CD4, in the studied group, a negative statistical significant correlation (r(48) = −0.350, *p* = 0.013) was found (Table 2).

The median value for CMV avidity for our patients was 64.40% (Table 1). The standard deviation was 20.5. A few (three patients) had undetectable CMV avidity (under 40% CMV avidity), another six patients had borderline avidity for CMV at the study moment and the rest of them (41 cases) had positive CMV avidity (values > 50% CMV avidity) (Table 1). Two patients with undetectable avidity to CMV had a good state of health in the moment of screening with CD4 over 200 cells/µL and no other complications or associated diseases. The third patient had a coinfection with HBV but without any symptoms or complications.

Avidity is defined as the aggregate strength with which a mixture of polyclonal IgG molecules binds to multiple antigenic epitopes on proteins. Avidity gradually matures over a few months. During the first months following primary infection, antibodies of the IgG type with low avidity are produced. Those produced by 4–6 months after infection have high avidity [9,10,11]. Since we did not measured the IgM CMV value in our study, we cannot state if the patients with undetectable or borderline (low) CMV avidity have recent infection or reinfection. CD4, as said before (Section 3.2), was over 200 cells/µL in the majority of cases. Only three patients had a CD4 value under 200 cells/µL and those patients were positive for IgG CMV and had a CMV avidity that is clearly positive. In the group of patients with CD4 > 200 cells/µL (47 patients), most of them (81% from patients with CD4 > 200 cells/µL) had a CMV avidity over 50% (positive avidity), six patients (representing 13% of the CD4 > 200 cells/µL patients) had borderline avidity and three patients (6% of the CD4 > 200 cells/µL patients) had an avidity under 40%.

CD4 value correlated negatively with IgG CMV values (r(48) = −0.350, *p* = 0.013) (Table 2) and also correlated negatively with CMV avidity (r(48) = −0.176) (Figure 3). This negative correlation may be the result of including patients in the seroconversion period in the study.

## 4. Discussion

In various parts of the world, depending on epidemiological factors, the seroprevalence of IgG CMV is 52.2–100% [12]. If we consider the initial lot of PLWH (Section 2.1), before applying the CMV-seropositivity inclusion criteria, we have in the South-East Dobrogea Region (Romania), Constanta County, a 71.42% prevalence of CMV infection in the PLWH feminine population. The analyzed population consists of Caucasian adults (between 18 years and 71 years) who are sexually active. The seroprevalence of IgG CMV in this group is higher than the seroprevalence of European countries [13] with a Caucasian population and a good economic situation where there are declared values around 40%, which is lower than the seroprevalence in developing afro-Indian counties with a black population and a lower economic status where seroprevalence is almost 100% [12,14].

Seroprevalence differences between countries may be explained by differences in the prevalence of key exposures related to CMV transmission: breastfeeding frequency and duration, crowding, childcare arrangements, and sexual behaviors. The economic conditions may produce the differences between other European countries and Romania because the population is Caucasian in all cases, so the race cannot be a factor of predisposition. If we measure the IgM CMV in IgG CMV-negative patients, the prevalence may be higher. In the vast majority of cases, CMV infection may be acquired in childhood, especially in countries where breastfeeding is largely practiced (South-East and East Europe, Africa, parts of Asia, and Latin America) and day-care institutions are crowded [13].

IgG CMV seroprevalence indicates that our patients harbor CMV virus in the latent form.

In a study on CD4 values of patients on ART, De La Mata et al. concluded that CD4 response to ART is modestly higher for those initiating ART in more recent years [15]. In our study, all patients received ART in the first years after HIV detection, but the CD4 level did not correlate with ART duration (*p* = 0.384). Another study from India [16] showed the same rise of CD4 values on ART therapy but the period of follow-up was only 6 months. In our study, patients had more than 1 year of ART. The longest period on ART was 22 years for our patients. Both cited studies did not appreciate the CMV infection status of their patients. CMV coinfection or immune characteristics of the patient may be the reason for the lack of correlation in present study.

In the current study, HIV viremia values correlated negatively with CD4 level (r(48) = −0.393, *p* = 0.005), a fact suggested by other studies that concluded a favorable evolution of HIV disease when the two variables are negatively correlated [8,17]. Berry’s research assessed the relationship between plasma HIV-2 RNA levels and CD4 lymphocyte percentage [17]. The study cohort was formed from 133 HIV-2 PLWH followed for 8 years. In his study, patients with undetectable HIV viremia had high levels of CD4. The cohort of Berry’s study had no coinfections declared. We had a 50-patient cohort, all HIV-CMV coinfected. In another study the test cohort was allmost symilar to our study cohort. The research was conducted on 40 women living with HIV and evaluated the CD level and HPV prevalence. Also, it had the same period of research (1 year) as our study, but they did not evaluate the CMV infection status of the patients. One of the study findings was that HIV viremia and CD4 are negatively correlated [8].

The level of viremia did not correlate with ART duration (*p* = 0.632) in our study. This may be because of lack of compliance to treatment or resistance to treatment as Liu et al. affirmed in their study on low levels viremia in patients under ART and resistance to treatment. The study of Liu was conducted on 6533 patients, under ART for more than 1 year, and stated clearly that lack of compliace to treatment determined high levels of HIV viremia even under ART [18].

In the paper of Ariyanto, the levels of CMV antibody increased on ART [19]. In our study, ART has no statistical correlation with IgG CMV values, but there is a positive correlation (r(48) = 0.024) between the two variables as stated in Ariyanto’s study as well. This might be due to small sample size, state of the HIV-CMV infection, the fact that we could not estimate the CMV infection moment and also the CMV infection duration until the detection moment. The only parameter we can appreciate is if the patients have CMV for more than 6 months (based on the fact that CMV avidity levels are clearly positive after six months from the infection). Also, a reactivation of CMV in HIV infection evolution is possible and in the absence of measured IgM CMV values, this cannot be appreciated. In another study held in North-Central Nigeria, the authors also found no statistical correlation between ART and IgG CMV values. The study had 360 PLWH with active and latent CMV infection rates, some of the patients being ART-naïve at the moment of the study [20].

In the study of Udeze et al. [20] CD4 values and IgG CMV values had no correlation. In our study, CD4 count and IgG CMV level correlated negatively (r(48) = −0.350, *p* = 0.013). The Nigerian study was developed with 360 patients with active and latent CMV infection; our study had 50 patients, all of them with latent CMV infection. Their study also included ART-naïve patients. In our study, all patients were on ART for more than 1 year. This may be the cause of the difference in level of correlation between the two variables.

Gomez-Mora et al. showed that increased IgG CMV responses were significantly correlated with a lower nadir and lower CD4+ T-cell counts, which is concordant with our study. The study of Gomez-Mora included 228 CMV-seropositive PLWH on ART (with a median of 10 years of therapy) and without a detectable HIV viral load [21]. We considered the threshold of CD4 value as 200 cells/µL (as limit for severe immunosuppression according to published studies [22] and WHO). In our study, in the patients with CD4 ≤ 200 cells/µL, the group average IgG CMV value was smaller than the average value in the CD4 > 200 cells/µL group of patients. Gomez-Mora’d study with a different threshold for CD4 obtained significantly higher CMV IgG levels among PLWH with poor immune recovery (CD4 < 350 cells/µL) subjects compared with adequate immune recovery (CD4 > 350 cells/µL) subjects [21]. This difference may come from the fact that the study of Gomez-Mora considered the threshold value CD4 = 350 cells/µL and we considered the threshold at 200 cells/µL, and the fact that in the present study, the CD4 ≤ 200 cells/µL group had only three patients. If we adjust the limit of CD4 to 350 cells/µL, values of IgG CMV in the group with CD4 < 350 cells/µL become higher than that in the group of CD4 > 350 cells/µL, which agrees with the study of Gomez-Mora. The number of patients in the CD4 < 350 cells/µL group of our study increases to 9 (out of 50 cases). These adjusted results sustain and confirm the idea of immunologic impairment of CMV infection in PLWH obtained in previous studies [23,24,25,26].

Booiman et al., in their study on CMV and HIV levels and differentiation of CD4 and CD8 lymphocytes, concluded that CMV avidity has higher values in PLWH and is suggestive of increased stimulation by CMV antigens, which may further contribute to T cell activation, exhaustion and terminal differentiation [26]. CMV infection was strongly associated with an increased proportion of terminally differentiated T cells and may explain the higher levels of this T cell type in PLWH within this study. Higher IgG CMV titers were associated with higher CD4+ T cell activation in the CMV-positive participants [26]. The study was conducted on 134 PLWH subjects and 79 non-infected controls, over 45 years of age with no detectable HIV viremia and with ART for more than 12 months. In the actual study, we found that CD4 value correlated negatively with CMV avidity (r(48) = −0.176). This may be because of the small number of patients, or the fact that, in this study, patients had ages ranging from 18 to 70 years old, they were only female and HIV viremia has heterogenic values. The majority of subjects had undetectable viremia (under 50 copies/mL) but there are a few with detectable viremia, which is different from Booiman’s lot characteristics. Also, CMV avidity has higher values in reinfection or reactivation of CMV [26]. Those situations may be associated with lower CD4 values as immunity is modified in PLWH. CD4+ lymphocytes participate in the antiviral effect. Cellular activation caused by CMV contributes to HIV pathogenesis by depletion of CD4+ T cells via apoptosis-induced cell death [27]. In PLWH, immunodepression has complex immune mechanisms, CD4 being only one of the multitude immune status indicators. The human immunodeficiency virus (HIV) infection produces a progressive weakening of the immune system by infecting and destroying cells involved in host defense, including CD4+ lymphocytes. A low level of CD4+ cells will allow opportunists like CMV infection to evolve into complications. Also, the evolution of CMV infection determines the lowering of CD4 levels and allows HIV to escape the immune system and evolve into AIDS and its complications. High values of immunoglobulins show a high level of interaction between immunologic cells and pathogens. A high value of CMV IgG and low level of CD4 cells may have future consequences on PLWH’s evolution and developing complications. CMV IgG seropositivity has been associated with immunosenescence and the development of cardiovascular diseases, cancer and other pathologies [28,29]. Patients with higher levels of CD4 and lower levels of CMV IgG are less predisposed to complications. Negative correlation between CD4 levels and IgG CMV or CMV avidity in our study show the good response to ART for evaluated patients because our lot is formed from patients with CD4 over 200 cells/µL, meaning that treatment keeps a good immunity status and the opportunistic infection under control.

Different coinfections, with the same mechanism of weakening of the immune system, were also reported in PLWH. These will exacerbate the symptoms and evolution to AIDS. The same negative correlation between CD4 levels and Immunoglobulins G type or level of viremia were present in opportunistic coinfection with other viruses like HBV or HCV [30,31] and parasites like *Leischmania* [32,33] or *Toxoplasma* [30]. In our study, PLWH CMV+ coinfected with HBV or HCV represent less than 20% of the study population. Most of them had CD4 over 200 cells/µL and no complications at the testing moment. Association between HIV seropositivity and CMV infection is recognized to be implicated in immune activation, senescence, cardiovascular complications, neurocognitive impairment along with a progressive loss of gut barrier function that leads to the translocation of microbial product that contributes to general inflammation and accelerated progression to AIDS and death [21,34,35,36]. In our lot of patients, there were no cases with cardiac complications, retinitis or other complications clearly determined by CMV infection (1 patient out of 50 had pneumopathies). The immune activation could not be appreciated in the absence of other specific investigations. The absence of the control group (hospital is dedicated to PLWH) made the evaluation of IgG CMV levels of PLWH CMV+ versus HIV seronegative, and CMV-seropositive patients impossible. It is recognized that levels of IgG CMV are higher in the first group of patients as a result of immune activation process. [21,26,37].

## 5. Conclusions

In this study, we analyzed IgG CMV levels and CMV avidity in 50 Romanian women living with HIV, as part of a larger study on CMV infection. We included in this study only patients exposed to ART more than 6 months. Viremia values had a negative correlation with CD4 level and no correlation with ART duration. Values of IgG CMV, compared with values of CD4+ cells, revealed a negative significant correlation that can be useful to clinicians in predicting possible complications in patients with HIV-CMV coinfection. This finding was also highlighted in other pathologies, and it may represent a useful marker for complications in immunodepressed patients. Moreover, it can be used as an indicator of disease outcome. Comparing IgG CMV values with the length of ART therapy, we identified a positive correlation between the two variables but with no statistical significance. No correlation was identified between CMV avidity and levels of HIV viremia, but CMV avidity correlated negatively with CD4+ cell value. This finding sets the stage for future research on a larger group. Since this research is based on a small, local study, with a limited number of cases, further studies are needed to achieve appropriate conclusions. The results of our study that show correlations between CD4+-positive-cell level and immunoglobulin level can be considered a starting point for future research directions, leading to the exploration of other inflammatory factors that may influence the relationship between other pathogens.

## Figures and Tables

**Figure 1 jpm-13-01539-f001:**
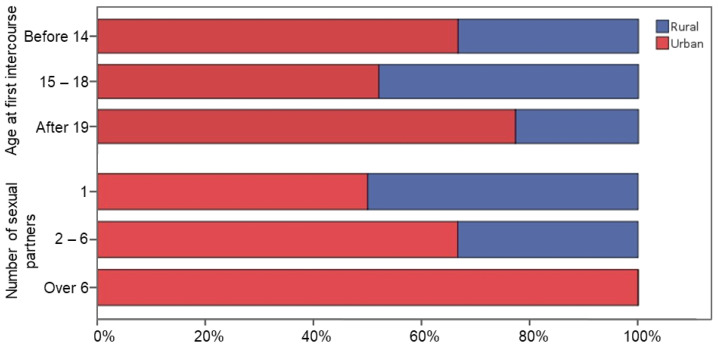
Age at sexual life debut and number of sexual partners declared by patients according to the rural or urban provenance.

**Figure 2 jpm-13-01539-f002:**
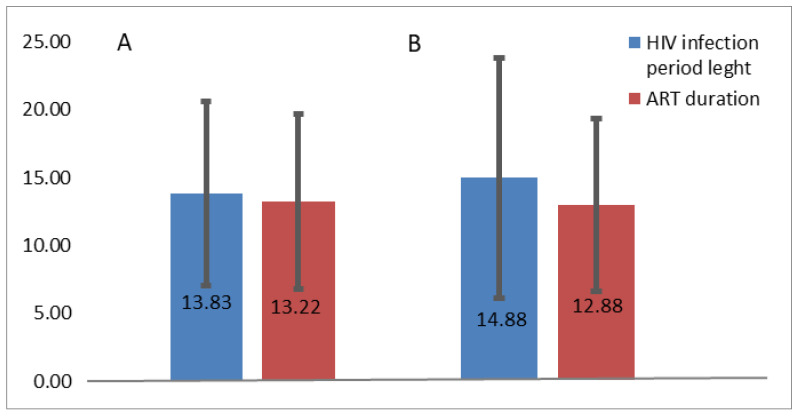
HIV infection period length and ART duration in rural and urban area: (**A**) Rural area—HIV infection and ART duration (SD values are 6.74 for HIV infection duration and 6.46 for ART duration). (**B**) Urban area—HIV infection and ART duration (SD values are 8.83 for HIV infection period length and 6.34 for ART duration).

**Figure 3 jpm-13-01539-f003:**
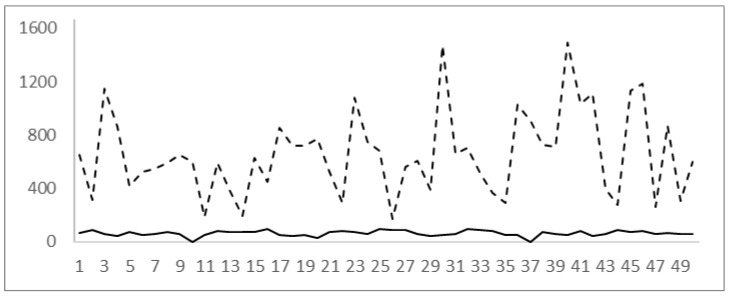
Percentage of CMV avidity (full line) and CD4 cells/microliters (dotted line) showing a negative correlation.

**Table 1 jpm-13-01539-t001:** Social and paraclinical characteristics of study cases.

Parameter	Social Data/Paraclinical Data	N/Value (SD)
Provenience area	Rural	18
	Urban	32
Patients’ first sexual intercourse age	Under 14 yo	3
	15–18 yo	25
	Over 19 yo	2
Sexual partner number	1 partner	10
	2–6 partners	39
	Over 6 partners	1
Median age at HIV diagnostic		18.82 (SD = 11.82)
HIV infection period length (in years)		14.5 (SD = 8.09)
HIV infection path	Parenteral	21
	Sexual	29
ART duration (median, in years)		13 (SD = 6.32)
HIV viremia (number of cases)	<50 copies/mL (undetectable)	41
	>50 copies/mL (detectable)	9
IgG CMV median value		215.71 (SD = 71.29)
CMV avidity median value		64.40 (SD = 20.95)
CMV avidity (number of cases)	<40% (negative)	3
	40–50% (borderline)	6
	>50% (positive)	41
CD4 median value		659.30 (SD = 318.71)
CD4 values (number of cases)	200 cells/µL or less	3
	Over 200 cells/µL	47

N = number of patients, SD = standard deviation.

**Table 2 jpm-13-01539-t002:** Spearman’s correlations between HIV viremia (copies/mL) and CD4 values (cells/µL), ART therapy (years) and IgG CMV values (U/mL), and CD 4 values and IgG CMV values (U/mL) performed on 50 patients. Significant correlations are marked with bold.

		rho	*p*
HIV viremia (copies/mL)	–CD4 (cells/µL)	**−0.393**	**0.005**
ART therapy (years)	–IgG CMV values (U/mL)	0.024	0.869
CD 4 values (cells/µL)	–IgG CMV values (U/mL)	**−0.350**	**0.013**

Correlations are significant at the 0.01 level (2-tailed).

## Data Availability

All presented in this study is contained within the article.

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
