# Peer review of "CMV and HIV Coinfection in Women from a Region in Eastern Europe"

_jpm, 2023, doi:10.3390/jpm13111539_

Round 1

Reviewer 1 Report

Comments and Suggestions for Authors

This manuscript discusses the prevalence of cytomegalovirus (CMV) infection prevalence among HIV-seropositive women in Southeastern Romania and its correlation with antiretroviral therapy (ART), CD4 levels, and the evolution of CMV disease. The study concludes CMV infection is prevalent among HIV-seropositive women in Southeastern Romania. The negative correlation between CD4 levels and IgG CMV values suggests a potential impact on immune function. 

I suggest the authors use the free software Grammarly since the manuscript has several grammatical and punctuation errors. Furthermore, the discussion should be more profound. I encourage authors to include the suggested references and more information that may strengthen the study.

-The authors claim in the abstract that ‘Fifty HIV seropositive women from Southeastern Romania were screened for CMV infection using antigen quantification’, but then in section 2.1. claim ‘Seventy HIV-seropositive women from Southeastern Romania (Dobrogea Region, Constanta County) were screened…’. Certainly, 50 were chosen, but the initial screening was 70.

-Figures 1 and 2 could be merged into one. Further, these graphs need a Y axis, not only the numbers but the actual vertical line.

-Avoid using these expressions ‘weak’ or ‘very weak’ for correlations. These undermine the findings. A correlation is a correlation that is demonstrated by statistical analysis and significance. ‘Weak’ or ‘strong’ has no meaning. 

-The discussion needs a better structure. The authors limited the mention of some previous studies and compared the previous data with the current findings in a superficial manner. The discussion could provide a more in-depth interpretation of the results and their clinical significance. Discuss the potential implications of the negative correlations between CD4 levels and IgG CMV values and the weak correlation between CMV avidity and CD4 values.

-To strengthen the discussion, consider comparing the study's findings with previous research on CMV and HIV co-infection. Discuss how your results align with or differ from existing literature.

-The conclusion section could be expanded to provide more concrete insights or recommendations based on the study's findings. What are the clinical implications, and what future research avenues should be explored?

-What has been shown in previous studies about the role of CMV infection in CD4 cells?

-The authors should mention that the negative correlation between CD4 levels and IgG values has also been demonstrated in other HIV co-infections such as co-infections of HIV and Leishmania, HCV, Toxoplasma, and HBV (DOI: 10.1016/j.heliyon.2023.e15055; DOI: 10.1002/iid3.794).

Comments on the Quality of English Language

The manuscript requires English editing.

Author Response

Thank you for your valuable comments and suggestions. We appreciate your help to publish a better manuscript and we tried to respond every suggestion or comment.

Reviewer 2 Report

Comments and Suggestions for Authors

The authors proposed an important issue to establish the prevalence of CMV infection among HIV seropositive women and also to evaluate and correlate antiretroviral therapy (ART) with CD4 level and CMV disease evolution. Their study found that most cases of CMV disease occurred in HIV-infected patients with advanced immunosuppression. The HIV-infected patients with cluster of differentiation 4 (CD4) T-cell counts <200 cells/μ l are at significant risk for CMV reactivation, leading to invasive disease. Their study provided valuable information for the prevention and administration of CMV infection, which can lead to serious health complications, among HIV-positive female populations.

Here are some questions and suggestions

1.The authors' description of the previous studies on HIV-positive population's CMV infection status and health impact is too brief. They are not the first to study this issue, and they need to provide a slightly more detailed summary of the previous studies reported in ref 3 - 6.

2.The authors mentioned that only 3 patients had CD4 value under 200 cells/µl and those patients were positive for IgG CMV and had CMV avidity clearly positive .  What was the general healthy condition of the three patientsDo they have any other opportunistic infections? The authors also mentioned that a few (3 patients) had CMV avidity undetectable. What was the general healthy condition and opportunistic infections about those patients?

3.The authors mentioned ART has no statistical correlation with IgG CMV values. They found a weak positive correlation (r(48)=0.024 ) between the two variables. Can the authors provide insights into the underlying mechanisms of this phenomenon?

Author Response

(The authors gave the same response as above.)
